# Combined BRCA2 and MAGEC3 Expression Predict Outcome in Advanced Ovarian Cancers

**DOI:** 10.3390/cancers14194724

**Published:** 2022-09-28

**Authors:** Emmanuel B. Omole, Iqbal Aijaz, James Ellegate, Emily Isenhart, Mohamed M. Desouki, Michalis Mastri, Kristen Humphrey, Emily M. Dougherty, Spencer R. Rosario, Kent L. Nastiuk, Joyce E. Ohm, Kevin H. Eng

**Affiliations:** 1Department of Cancer Genetics and Genomics, Roswell Park Comprehensive Cancer Center, Buffalo, NY 14263, USA; 2Department of Biostatistics and Bioinformatics, Roswell Park Comprehensive Cancer Center, Buffalo, NY 14263, USA; 3Department of Pathology, Roswell Park Comprehensive Cancer Center, Buffalo, NY 14263, USA; 4Department of Pharmacology and Therapeutics, Roswell Park Comprehensive Cancer Center, Buffalo, NY 14263, USA; 5Department of Urology, Roswell Park Comprehensive Cancer Center, Buffalo, NY 14263, USA

**Keywords:** ovarian cancer, immunohistochemistry, biomarker, MAGEC3, BRCA2

## Abstract

**Simple Summary:**

Early detection of ovarian cancer is a significant clinical challenge, with most women presenting with advanced stages of disease at initial diagnosis. The aim of this study was to evaluate the role of MAGEC3 and BRCA2 in epithelial ovarian cancer progression. We evaluated the effect of MAGEC3 and BRCA2 on the other’s expression. We tested this in humans using immunohistochemical staining of human tumor samples obtained from patients with epithelial ovarian cancer (*n* = 357). We found a weak inverse correlation between MAGEC3 and BRCA2 expression in epithelial ovarian cancers. Further, our data suggest that the combined expression of MAGEC3 and BRCA2 may be a better predictor of outcomes in patients than the individual markers alone.

**Abstract:**

Like *BRCA2*, *MAGEC3* is an ovarian cancer predisposition gene that has been shown to have prognostic significance in ovarian cancer patients. Despite the clinical significance of each gene, no studies have been conducted to assess the clinical significance of their combined expression. We therefore sought to determine the relationship between MAGEC3 and BRCA2 expression in ovarian cancer and their association with patient characteristics and outcomes. Immunohistochemical staining was quantitated on tumor microarrays of human tumor samples obtained from 357 patients with epithelial ovarian cancer to ascertain BRCA2 expression levels. In conjunction with our previously published MAGEC3 expression data, we observed a weak inverse correlation of MAGEC3 with BRCA2 expression (r = −0.15; *p* < 0.05) in cases with full-length BRCA2. Patients with optimal cytoreduction, loss of MAGEC3, and detectable BRCA2 expression had better overall (median OS: 127.9 vs. 65.3 months, *p* = 0.035) and progression-free (median PFS: 85.3 vs. 18.8 months, *p* = 0.002) survival compared to patients that were BRCA2 expressors with MAGEC3 normal levels. Our results suggest that combined expression of MAGEC3 and BRCA2 serves as a better predictor of prognosis than each marker alone.

## 1. Introduction

Ovarian cancer is a heterogeneous disease, and remains the leading cause of mortality from gynecologic malignancies, with more than 19,000 new cases and 12,810 deaths expected in the United States in 2022 [1]. Epithelial ovarian cancer (EOC), which accounts for over 90% of all ovarian cancers, is comprised of five distinct subtypes, amongst which high-grade serous ovarian carcinoma (HGSOC) is the most commonly diagnosed [2]. HGSOC is also the most lethal subtype, accounting for about 70% of all epithelial ovarian cancer cases [3,4] and 70–80% of all ovarian cancer deaths [5] due to it being diagnosed at an advanced stage [6].

Since most women present with advanced stages of disease [4,7,8], early detection of ovarian cancer remains an increasingly important clinical challenge [2,8,9]. There is a compelling argument for novel ovarian cancer screening methodologies, given that the current strategy results in the early detection of ovarian cancer in only ~20% of patients [10]. Advances in ovarian cancer diagnosis include the use of genetic testing to identify mutated genes associated with epithelial ovarian cancer in patients, with a known family history of breast and ovarian cancer, such as *TP53*, *PIK3CA*, *KRAS*, and *BRCA1/2* [2,8].

*BRCA1/2* genes encode proteins involved in the repair of DNA double-strand breaks via the homologous recombination pathway [11,12,13,14]. Germline and somatic mutations, as well as promoter methylation, are the common causes of *BRCA1/2* dysfunction [15,16] which impairs DNA repair leading to an increased risk of tumorigenesis [17]. *BRCA2* mutations specifically are associated with a 15–30% lifetime risk of ovarian cancer [18]. Most of the deleterious mutations that have been identified in ovarian cancer families are frameshift (insertions or deletions) mutations or nonsense mutations that result in premature termination of the protein-coding sequences [19,20,21,22]. Resultant BRCA2 mutant proteins retain their N-terminal transactivation domain which enables its recombinational repair and checkpoint roles [23,24,25] but the C-terminal, which harbors the nuclear localization signals (NLS), is lost [25]. The inability to translocate into the nucleus explains why these mutants are nonfunctional and unable to facilitate the DNA repair process in most cases [20,25,26].

*BRCA1/2* mutations serve as a marker of vulnerability to PARP inhibitors (olaparib, rucaparib, and niraparib). These have become viable treatment options for patients with advanced epithelial ovarian cancer, due to enhanced clinical effectiveness and tolerable toxicity [27,28,29,30,31]. While women with *BRCA1/2* mutations retain the best response to PARP inhibitors [32], tumors with genomic instability still show a clinically significant response to therapy [33], prompting researchers to consider whether other markers such as ATM, FANC A/F, CHK2, RAD51B/C, and CDK12 could be used as de facto surrogate markers of deficient homologous repair [34,35] to expand the pool of women who may benefit from PARP inhibitor therapy.

One such potential marker is MAGEC3. MAGEC3 is a member of the “Type II” melanoma antigen gene (MAGE) family by DNA sequence homology and has low but ubiquitous protein expression in most tissues with high expression in testes and tonsils [36]. In the context of cancer, members of the family have shown varying functions in DNA repair and cell cycle regulation, [37,38] as well as the potential application as markers of stemness in cancer cells [39]. Biochemically, MAGE proteins bind specifically to E3 RING ubiquitin ligases through their MAGE homology domains (MHDs) and regulate protein ubiquitination by enhancing E3 ligase activity [37,38,40,41]. Previously, we described a linkage between early-onset ovarian cancers [42] and MAGEC3 [43,44] and more recently found that MAGEC3 is a potential prognostic biomarker in ovarian cancer, as loss levels were associated with a favorable progression-free survival (PFS) in ovarian cancer patients [36].

The relationship between both BRCA2 and MAGEC3 protein expression and their combined clinical significance in ovarian carcinoma is unknown. Immunohistochemistry (IHC) is a low-cost and easily accessible technique that allows for the query of both N- and C-terminal domains of the BRCA2 protein, including protein truncation of the C-terminal of the BRCA2 protein which severely diminishes protein function [45]. Similarly, IHC has also been used to detect the expression of the MAGEC3 protein [36,46]. Herein, we describe the relationship between MAGEC3 and BRCA2 in patient samples using IHC and tested the hypothesis that their co-expression is correlated with clinical outcomes.

## 2. Materials and Methods

### 2.1. Ovarian Cancer Patients

Two cohorts of patients were used: (1) Patients with advanced high-grade serous ovarian carcinoma treated at Roswell Park Cancer Institute (RPCI) who had previously had BRCA2 genetic testing. This cohort was used for the pilot experiment and classified into three categories based on the results of the genetic testing: BRCA2-Wildtype, BRCA2-Mutant, and BRCA2-Loss. (2) Patients with a primary diagnosis of ovarian cancer treated at RPCI by maximal debulking surgery and first-line platinum-based chemotherapy between 2000 and 2012 were obtained from a curated database under an institutional review board-approved protocol. Tumor blocks from these cohorts were taken during primary debulking surgery and prior to the initiation of systemic therapy.

In total, there were *n* = 3 ovarian cancer patients from cohort 1 with sequenced tumor samples processed as whole tissue sections for the pilot experiment and *n* = 357 ovarian cancer patients from cohort 2 with un-sequenced tumor samples for the tissue microarray analysis. The detailed handling protocol for the tissue specimens has been previously described [47].

### 2.2. Whole Tissue Sections

The whole tissue sections were derived from formalin-fixed and paraffin-embedded tumor tissues obtained from three sets of sequenced ovarian tumor cases in cohort 1 to qualify antibody reactivity. IHC H-scores were generated using *n* = 30 photographs of randomly selected fields on each slide.

### 2.3. Tissue Microarray (TMA) Construction

The tissue microarrays (TMA) were constructed using formalin-fixed and paraffin-embedded tumor tissues punched from each donor block from cohort 2. Triplicate TMAs (containing three representative cores from each tumor) were prepared and stained to assess tumor heterogeneity. These TMAs included all epithelial ovarian cancer subtypes, non-tumor control tissues, and reference cores for normalization across different blocks.

### 2.4. Antibody Validation

The Human Protein Atlas [48] (HPA052067) was used to vet the BRCA2 antibodies, in accordance with the International Working Group for Antibody Validation [49], and were evaluated and authorized for IHC and supported for antigen specificity. The BRCA2 antibodies are directed against the BRCA2 C-terminal amino acids (2587–2601 amino acids, Abcam ab53887, Cambridge, MA, USA) and BRCA2 N-terminal amino acids (100–150 amino acids, Proteintech 19791-1-AP, Rosemont, IL, USA).

The MAGEC3 antibody has been previously described [36].

### 2.5. Immunohistochemistry

Formalin-fixed paraffin sections were cut at a thickness of 4μm, placed on glass slides, and heated at 60 °C for 1 h. The TMA sections were then deparaffinized with xylene and ethanol and rinsed in water. For antigen retrieval, slides were pretreated in an antigen retrieval high pH solution (catalog #GV804, Dako, Santa Clara, CA, USA) for 20 min and allowed to cool for 20 min at room temperature. The slides were incubated with a solution of 3% H2O2 for 10 min. “BRCA2 N-terminal” antibody (catalog #19791-1-AP, Proteintech, Rosemont, IL, USA) was added to one set of slides at 1:500 dilution while the “BRCA2 C-terminal” antibody (catalog# ab53887, Abcam, Cambridge, MA, USA) was added to the other set of slides at 1:500 dilution and incubated for 60 min at room temperature. This was followed by incubation with biotinylated goat anti-rabbit secondary antibody (catalog # 14708, Cell Signaling Technology, Boston, MA, USA), 1:1000 dilution for 15 min at room temperature. ABC reagent (catalog #PK 6100, Vector Labs, Burlingame, CA, USA) was used for signal enhancement and applied for 30 min. Slides were incubated for 5 min with 3,3′-diaminobenzidine (DAB) substrate (catalog #K3467, Dako, Carpinteria, CA, USA) and then counterstained for 20 s with DAKO Hematoxylin (catalog #CS700, Dako, Carpinteria, CA, USA). Slides were dehydrated in several baths of graded alcohols and xylenes before being cover-slipped.

TMAs were previously stained for MAGEC3 as described [36].

### 2.6. Digital Image Analysis

Digital images of both IHC-stained whole tissue sections and IHC-stained TMA slides were captured using a whole slide scanner at 10× magnification (Aperio Scanscope, Leica Microsystems, Milton Keynes, UK), and the digital images were stored in SVS format. The digital images were retrieved using a file management web interface (Aperio eSlide Manager, version 12.4.3.8007, Leica Biosystems, IL, USA) and reviewed with the server software (Aperio Imagescope, version 12.3.3.5048, Leica Biosystems, IL, USA). Digital images of representative whole tissue section fields and TMA cores were extracted through the server software (Aperio Imagescope, version 12.3.3.5048, Leica Biosystems, IL, USA) at 10× magnification and quantified into H-scores using an automated IHC profiler built in a digital image analysis software (ImageJ version 1.52a, National Institute of Health, Bethesda, MD, USA) [50]. 

### 2.7. Statistical Methods

After the IHC scoring was complete, the optimal cutpoint of the IHC staining score for BRCA2 expression was determined using H-scores obtained for the C-terminal of the sequenced ovarian tumor cases with the aid of the R package, cutpointr [51]. We stratified the BRCA2 C-terminal H-scores into two different groups (BRCA2 expressors and BRCA2 non-expressors).

All statistical analyses were performed using the R 3.1.2 statistical computing language. Associations between BRCA2/MAGEC3 expression and categorical variables were tested by chi-square tests and continuous variables were tested by one-way ANOVA tests. Survival probabilities were calculated by Kaplan–Meier analysis using log-rank testing. A nominal significance threshold of 0.05 was used unless otherwise specified. The multivariate analysis included stage, categorized as early (I, II, IIIA, or IIIB) or late (IIIC or IV), grade (1 vs. 2/3), debulking status (optimal vs. suboptimal), and platinum-sensitive versus refractory disease.

## 3. Results

### 3.1. Validation of IHC Measurements of BRCA2 Protein Expression in Cancer Tissue

We first tested whether BRCA2 protein expression confirmed the predicted expression by molecularly characterized and sequenced ovarian tumor samples, via IHC. The adopted two-antibody IHC strategy, which utilizes antibodies specific to the N-terminus and C-terminus, allows for discrimination between the presence of normal protein, protein truncation, and loss of protein expression in the tumor samples (Figure 1A). These antibodies were applied to three cases confirmed to have BRCA2 wild type (WT), BRCA2 truncating mutation in exon 11 (Mutant), and total loss of BRCA2 (Loss) through sequencing. Visual analysis of IHC staining shows that, as expected, wildtype samples have adequate staining observed with both the N-terminal and C-terminal antibodies, mutant samples have staining only at the N-terminal and not the C-terminal, and BRCA2 loss has poor staining intensity seen with both antibodies (Figure 1B).

To further validate this strategy, the staining intensity and percentage of positive tumor cells were quantified using an automated IHC H-scoring algorithm built in ImageJ. The resulting H-scores show that N-terminal expression levels are significantly different between BRCA2 wild type, BRCA2 mutant, and BRCA2 loss in sequenced cases (WT vs. Mutant: *t*-test, *p* < 0.001; WT vs. Loss: *p* < 0.001; Mutant vs. Loss: *p* < 0.001) (Appendix A). However, given that a truncating mutation renders the protein non-functional, samples that demonstrate that a lack of C-terminal staining should functionally mimic BRCA2 loss. The difference in C-terminal expression levels, when BRCA2 wildtype was compared to BRCA2 mutant and BRCA2 loss cases, was observed to be statistically significant (WT vs. Mutant: *t*-test, *p* < 0.001; WT vs. Loss: *p* < 0.001), while the difference in C-terminal expression levels for the BRCA2 mutant and BRCA2 loss cases was not statistically significant (Mutant vs. Loss: *t*-test, *p* = 0.9) (Figure 2A). Given its ability to differentiate wild-type cases from the functionally impaired mutant and loss cases, C-terminal scoring was used for further analyses. C-terminal scores for the BRCA2 mutant and BRCA2 loss cases were combined and termed “BRCA2 non-expressors” while the BRCA2 wild-type cases were termed “BRCA2 expressors.” When the same IHC strategy was applied to 357 un-sequenced patient samples with unknown BRCA2 status, we used the C-terminal expression levels of the BRCA2 expressors and non-expressors to determine an optimal cutoff point for stratifying the un-sequenced cases. The optimal cutoff point utilized to stratify the larger un-sequenced cohort of patients into “BRCA2 expressors” and “BRCA2 non-expressors” was an H-score of 49.5 (Figure 2B).

MAGEC3 protein expression quantitation by IHC for patients in this cohort was previously reported in a recent publication from our lab [36]. Expression levels were dichotomized at the median into either “MAGEC3 Normal” or “MAGEC3 Loss” based on the findings that higher levels of MAGEC3 were not significantly different from expression in normal ovary tissue (*t*-test, *p* = 0.368), while lower levels of MAGEC3 were significantly lower (*t*-test, *p* < 0.001). Given our interest in the combined effect of MAGEC3 and BRCA2 expression levels on patient prognosis, we used MAGEC3 expression data from the prior publication as well as the BRCA2 levels ascertained in this study for subsequent analyses.

### 3.2. Clinical Characteristics of Ovarian Cancer Microarray Patient Population

A total of 357 tumor tissues from patients with ovarian and primary peritoneal cancer were analyzed using IHC. From the demographic information of de-identified patients in this study (Table 1), the mean age of diagnosis was 63 years (range 21–93 years) and 52.1% of patients were diagnosed after 2006. Most of the patients were white (334; 94.9%), non-Hispanic (350; 99.4%), and presented with advanced/late-stage disease (81.5%), poorly differentiated tumors (73.5%) and serous histology (79.6%). Platinum-sensitive disease was demonstrated in 156 of the 357 patients (55.5%) with 125 patients classified as platinum-resistant or treatment-refractory (42.5%). For survival data, the patients were followed for an average of 65.2 months (maximum 274.2 months).

Using the optimal cutoff point calculated above, 299/257 (83.8%) of the patients were classified as BRCA2 expressors while 58/357 (16.2%) were BRCA2 non-expressors. Using the previously reported stratification of MAGEC3, 173/357 (48.5%) of patients in this cohort are MAGEC3 loss while 184/357 (51.5%) are MAGEC3 normal. When considering the combination of BRCA2 and MAGEC3 levels, 34 (9.5%) were BRCA2 non-expressors and showed MAGEC3 loss levels, 139 (40%) were BRCA2 expressors with MAGEC3 loss levels, 24 (6.7%) were BRCA2 non-expressors with MAGEC3 normal levels, and 160 (44.8%) were BRCA2 expressors with MAGEC3 normal levels. Through extensive statistical analysis, BRCA2 expression and normal MAGEC3 levels were associated with cases ascertained after 2006 (Chi-sq test, *p* < 0.001), ovary as the primary site of origin (Chi-sq test, *p* < 0.001), poorly/undifferentiated tumors (Chi-sq test, *p* = 0.03) and serous histology (Chi-sq test, *p* < 0.001).

### 3.3. BRCA2 Protein Shows a Weak Negative Correlation with MAGEC3 Expression in Ovarian Cancer Tumor Samples

The H-scores from the semi-quantitative analysis of MAGEC3 TMAs in our previous study [36] were correlated with the H-scores of BRCA2 TMAs in this current study using a two-sided Pearson’s correlation test. The initial assessment of the relationship between MAGEC3 and BRCA2 expression in the tumor samples revealed no correlation (Figure 3A). However, when the cases were stratified based on BRCA2 expression, the MAGEC3 H-scores were shown to be inversely correlated with the H-scores of BRCA2 expressors in the ovarian tumor samples (r = −0.15; *p* < 0.05) (Figure 3B), while a positive correlation was observed in the BRCA2 non-expressors (r = 0.32; *p* < 0.05) (Figure 3C).

### 3.4. MAGEC3 and BRCA2 Association with Prognosis in Epithelial Ovarian Cancer

The impacts of age at diagnosis, disease stage, grade, histology, cytoreduction status, and MAGEC3/BRCA2 expression on clinical outcome were analyzed using both univariate and multivariate Cox proportional hazards regression analyses (Table 2). The age at diagnosis, grade, and R0 cytoreduction were univariately associated with prognosis in epithelial ovarian cancers. Multivariate regression analysis indicated that patients with advanced stage disease (HR = 3.28, 95% CI 2.05–5.26, *p* < 0.001) and poorly/undifferentiated tumors (HR = 1.34, 95% CI 1.02–1.76, *p* = 0.04) were statistically significant predictors of survival.

Independent evaluation of the relationship between BRCA2 expression and patient outcomes in our cohort of advanced ovarian cancer cases revealed that BRCA2 expression levels did not impact overall survival (log-rank *p* = 0.198) and progression-free survival (log-rank *p* = 0.15) in patients with optimal cytoreduction (Appendix A), whereas independent evaluation of MAGEC3 expression in our previous study revealed that women with MAGEC3 loss had better progression-free survival (log-rank *p* = 0.002) [36]. In patients with optimal cytoreduction, subgroup analyses of combined expression of MAGEC3 and BRCA2 showed that patients that were BRCA2 expressors with MAGEC3 loss levels had better overall survival compared to patients that were BRCA2 expressors with MAGEC3 normal levels (median OS: 127.9 vs. 65.3 months; log-rank *p* = 0.035) (Figure 4A). This was also observed for progression-free survival (median PFS: 85.3 vs. 18.8 months; log-rank *p* = 0.002) (Figure 4B). Conversely, in BRCA2 non-expressor cases, having MAGEC3 loss levels conferred no survival benefit over MAGEC3 normal levels in overall survival (log-rank *p* = 0.677) nor progression-free survival (log-rank *p* = 0.753) (Figure 4C,D).

## 4. Discussion

This study aimed to evaluate BRCA2 and MAGEC3 for their influence on epithelial ovarian cancer progression and to assess the clinical significance of their combined expression. Like BRCA2 [52], MAGEC3 protein expression has been identified for evaluation as a novel prognostic biomarker in ovarian cancer [36]. However, the relationship between both BRCA2 and MAGEC3 protein expression and their combined clinical significance in ovarian carcinoma is unknown.

The growing importance of IHC as a tool for detecting protein dysfunction via protein expression profiles, combined with its ease of use and practicality in identifying the protein of interest within the tumor’s cellular compartment, makes it a highly appealing technique to measure validated biomarkers. It proved to be a viable and cost-effective approach for determining BRCA2 protein expression in tumor samples for this study, as similarly reported by other studies [53,54], as well as for determining MAGEC3 protein expression [36]. We used a dual antibody strategy to discriminate differential expression of BRCA2 at both the N-terminal and C-terminal. Because of its ability to differentiate the wild-type case from mutant and loss cases, C-terminal expression was used to define BRCA2 expression in a large un-sequenced cohort. Prior studies also support the idea that the C-terminal antibody could be useful in screening cancers for BRCA mutations as well as BRCA2 protein expression in patients with unknown mutation status [45,53,55]. Using this approach, we identified 83.8% (299/357) of patients as BRCA2 expressors and 16.2% (58/357) as BRCA2 non-expressors.

In the assessment of the relationship between MAGEC3 and BRCA2 proteins in the tumor samples from the un-sequenced cohort, there was no significant correlation of protein expression when looking at all cases. However, when we looked at only BRCA2 expressors, there was an inverse correlation with MAGEC3 expression. An interesting conjecture emerges that, like other proteins in the MAGE family, MAGEC3 is interacting with an E3 ligase which leads to the ubiquitination and subsequent degradation of a protein, in this case, BRCA2. Because this downregulation was only observed in BRCA2 expressors, it is possible that it may only occur in cells expressing a full-length BRCA2 protein. Preliminary in vitro data from our lab supports this hypothesis, as overexpression of MAGEC3 results in a reduction in BRCA2 protein level in both fibrosarcoma (HT1080) and ovarian cancer (SKOV3) cell lines [56]. It is important to note that both the HT1080 and SKOV3 cell lines were reported to have a wild-type *BRCA2* gene [57].

In the unsequenced cohort of patients, survival analyses for overall and progression-free survival showed no significant relationship between BRCA2 expression and patient outcomes; whereas our previous study of the same cohort showed that women with MAGEC3 loss had better progression-free survival [36]. For ovarian cancer patients with optimal cytoreduction, combined expression of MAGEC3 and BRCA2 showed that patients that were BRCA2 expressors with MAGEC3 loss levels had better overall and progression-free survival compared with patients who were BRCA2 expressors with normal MAGEC3 levels. Interestingly, in patients that were BRCA2 non-expressors, MAGEC3 loss levels conferred no survival advantage over MAGEC3 normal levels for both overall survival and progression-free survival. Given our results showing that MAGEC3 downregulates BRCA2, we anticipated that higher levels of MAGEC3 would mimic BRCA2 loss status resulting in susceptibility to platinum-based therapies and prolonged survival. However, we note that the correlation observed in these patients was weak and that the in vitro experiments that showed downregulation of BRCA2 were performed at supraphysiological levels of MAGEC3, which would enhance its downregulating effects. A possible explanation for the trends observed is likely related to MAGEC3′s additional role in augmenting DNA repair [36]. Additional unpublished data from our lab revealed that ovarian cancer cells expressing MAGEC3 fare better under cisplatin (CDDP) insult by clearing cisplatin adducts [56]. These results led us to conclude that MAGEC3 is a prognostic marker in cases that are BRCA2 expressors and that its role in DNA repair, as opposed to its weaker role in downregulating BRCA2, is responsible for the trends observed. Because MAGEC3 only retains its prognostic ability in BRCA2 expressors, the combined expression remains a better predictor of outcomes in patients than the individual markers alone.

The limitations of our study include the use of only three sequenced ovarian tumor cases with known genomic status to develop our IHC assay, as well as the use of retrospective cohorts for the ovarian cancer TMAs. To reduce batch-processing effects and resulting slide-to-slide variation, the scores were normalized using normal control cores. Slide staining was performed by the pathology core facility at Roswell Park Comprehensive Cancer Center using appropriate antibodies for IHC.

Despite these limitations, our study has considerable strengths, including the analysis of a large cohort of patients to provide insight into the relationship between MAGEC3 and BRCA2 protein expression in ovarian cancer cases. Additionally, the integration and utilization of the automated semi-quantitative IHC profiler built into Image J for assessment of the expression levels of the proteins eliminated any visual bias and high-level observer variation that might have occurred with conventional and manual pathological TMA slide assessment. This was reported as effective in similar studies [58,59].

## 5. Conclusions

The combined expression of MAGEC3 and BRCA2 serves as a better predictor of prognosis than either marker alone, as MAGEC3 only retains its prognostic significance in BRCA2 expressors. Ovarian cancer cases expressing BRCA2 with low levels of MAGEC3 fare better than those with normal levels. This is contrary to expectations, given MAGEC3′s role in downregulating BRCA2. Given the role of MAGEC3 in augmenting DNA repair, we propose that the loss of this augmentation explains the favorable trend of patients with low MAGEC3 levels. Further work is needed to confirm this hypothesis.

## Figures and Tables

**Figure 1 cancers-14-04724-f001:**
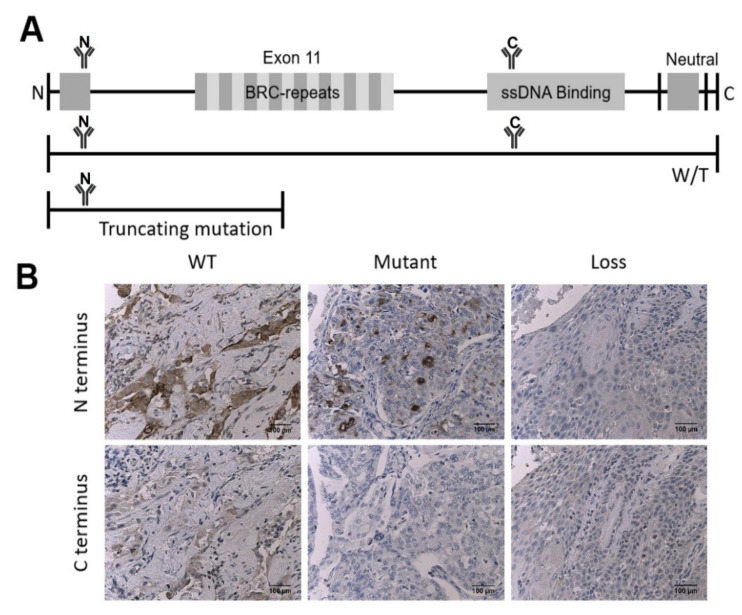
Schematic showing immunohistochemical staining of BRCA2 protein using two-antibody strategy. (**A**) A full-length gene model, a full-length wild-type protein model bound by both the N-terminal and C-terminal antibodies, and a truncated protein bound by only the N-terminal antibody (loss protein model not shown as it is not expressed and therefore cannot be bound by either antibody). (**B**) Representative microscopy images of sequenced ovarian cases with known genomic status, magnification 100×, scale bar = 100 µm. Wild-type (WT) expression pattern with staining of strong intensity with N-terminal antibody (top left) and moderate intensity with C-terminal antibody (bottom left) in some of the tumor cell nuclei; aberrant expression with moderate staining intensity with N-terminal antibody (top middle) and poor staining intensity with C-terminal antibody (bottom middle) in the verified case with truncating mutation (Mutant); poor staining intensity of both N-terminal (top right) and C-terminal (bottom right) antibodies in the verified case with BRCA2 loss (Loss).

**Figure 2 cancers-14-04724-f002:**
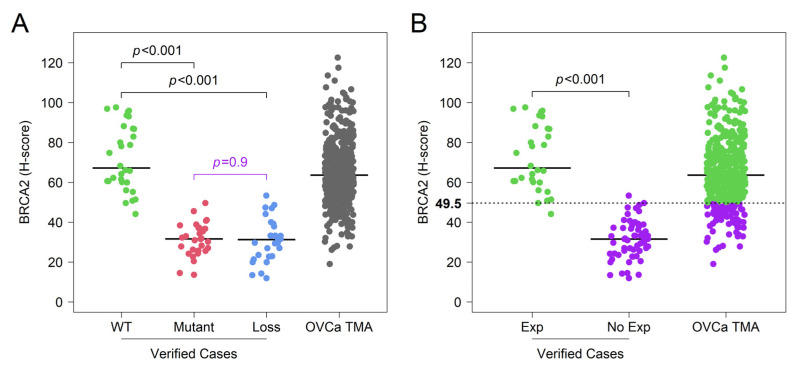
BRCA2 protein expression in the sequenced ovarian tumor cases and un-sequenced TMAs. (**A**) C-terminal expression of the BRCA2 wildtype (WT) is statistically different compared to BRCA2 mutant (Mutant) and BRCA2 loss (Loss) in the sequenced ovarian tumor cases, but there is no statistical difference between the Mutant and Loss cases. This was plotted alongside the distribution of un-sequenced TMA H-scores. (**B**) For the sequenced ovarian tumor cases, BRCA2 C-terminal expression of the WT was categorized as BRCA2 expressors (Exp) while those of the Mutant and Loss were grouped as BRCA2 non-expressors (No Exp). An optimal cutoff point of 49.5 was determined based on the C-terminal expression levels of the sequenced ovarian tumor cases and was used to stratify the un-sequenced ovarian cancer TMA (OVCa TMA) H-scores into Exp and No Exp.

**Figure 3 cancers-14-04724-f003:**
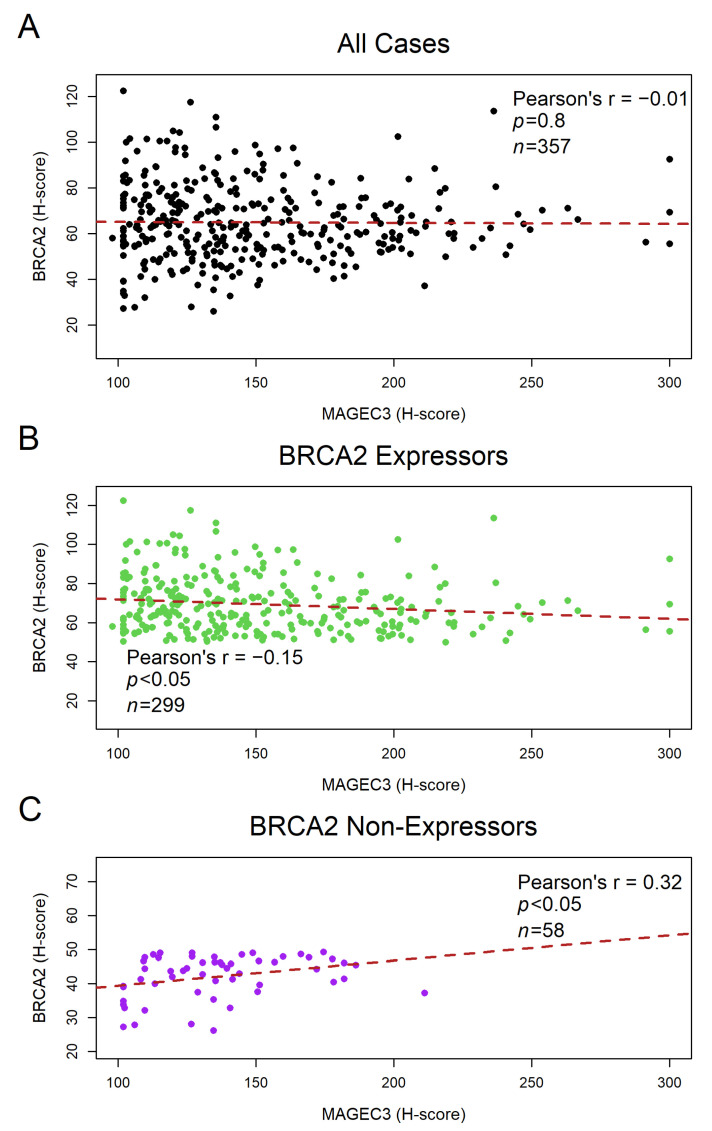
Correlation between MAGEC3 protein and BRCA2 C-terminal protein expression in ovarian tumor TMAs. (**A**) No correlation between MAGEC3 and BRCA2 expression in the tumor samples prior to stratification based on BRCA2 expression levels. (**B**) MAGEC3 H-scores were observed to be inversely correlated with the H-scores of BRCA2 expressors. (**C**) MAGEC3 H-scores were observed to be positively correlated with the H-scores of BRCA2 non-expressors.

**Figure 4 cancers-14-04724-f004:**
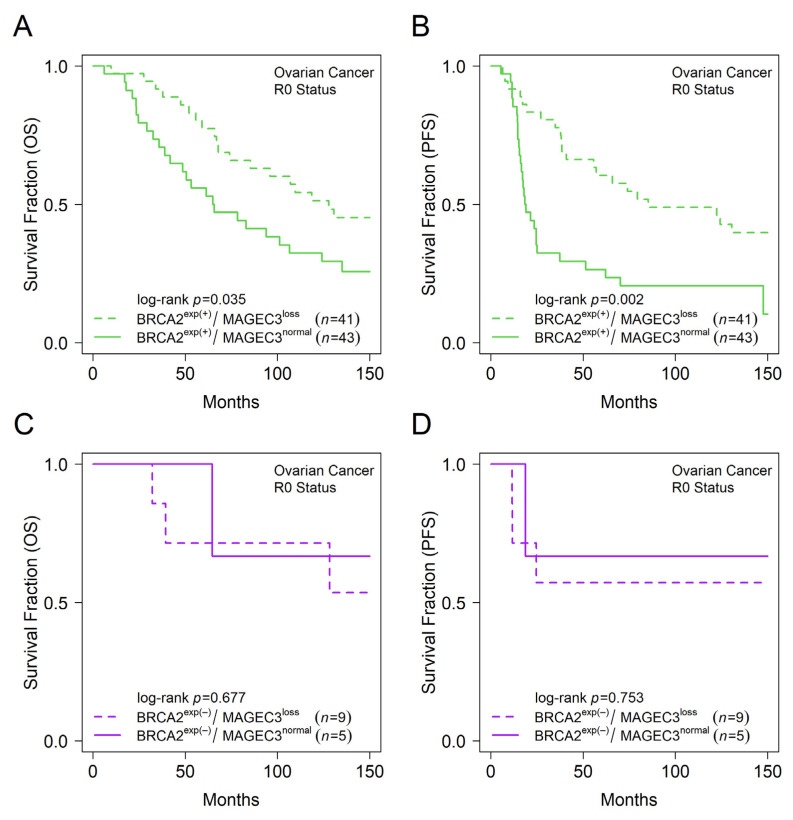
Expression of MAGEC3 and BRCA2 predict outcomes in ovarian cancer patients with optimal cytoreduction. Kaplan–Meier plot showing (**A**) overall survival trends for MAGEC3 normal and loss cases in BRCA2 expressors, (**B**) progression-free survival trends for MAGEC3 normal and loss cases in BRCA2 expressors, (**C**) overall survival trends for MAGEC3 normal and loss cases in BRCA2 non-expressors, (**D**) progression-free survival trends for MAGEC3 normal and loss cases in BRCA2 non-expressors for ovarian cancer patients with optimal cytoreduction.

**Table 1 cancers-14-04724-t001:** Clinical characteristics of the ovary discovery cohort by MAGEC3 and BRCA2 protein expression level.

Characteristic		MAGE^loss^		MAGE^normal^		
	All Ovary Patients	BRCA2^exp(−)^	BRCA2^exp(+)^	BRCA2^exp(−)^	BRCA2^exp(+)^	*p*-Value
	*n* = 357	*n* = 34	*n* = 139	*n* = 24	*n* = 160	
Age of Diagnosis (years)						0.1 ^†^
Mean (Range)	63 (21–93)	61 (21–85)	63 (31–93)	69 (47–89)	63 (21–89)	
Missing	0	0	0	0	0	
Year of Diagnosis (*n* (%))						<0.001 ^‡^
Before 2006	177 (52.1)	14 (43.8)	104 (81.9)	5 (21.7)	54 (34.2)	
After 2006	163 (47.9)	18 (56.3)	23 (18.1)	18 (78.3)	104 (65.8)	
Missing	17	2	12	1	2	
Race (*n* (%))						
White	334 (94.9)	31 (94)	130 (96.3)	24 (100)	149 (93.1)	0.2 ^§^
Black or African American	8 (2.3)	1 (3)	0 (0)	0 (0)	7 (4.4)	
Other	10 (2.8)	1 (3)	5 (3.7)	0 (0)	4 (2.5)	
Missing	5	1	4	0	0	
Hispanic (*n* (%))						
No	350 (99.4)	34 (100)	133 (99.3)	24 (100)	159 (99.4)	0.9 ^§^
Yes	2 (0.6)	0 (0)	1 (0.7)	0 (0)	1 (0.6)	
Missing	5	0	5	0	0	
Primary (*n* (%))						
Ovary	310 (88.1)	27 (81.8)	131 (94.9)	16 (66.7)	136 (86.6)	<0.001 ^‡^
Primary Peritoneal	42 (11.9)	6 (18.2)	7 (5.1)	8 (33.3)	21 (13.4)	
Missing	5	1	1	0	3	
FIGO Stage (*n* (%))						0.6 ^‡^
I/II/IIIA/B	65 (18.5)	8 (24.2)	28 (20.6)	4 (17.4)	25 (15.7)	
IIIC/IV	286 (81.5)	25 (75.8)	108 (79.4)	19 (82.6)	134 (84.3)	
Missing	6	1	3	1	1	
Grade (*n* (%))						0.03 ^‡^
Well/Moderately differentiated	93 (26.5)	5 (14.7)	44 (32.6)	2 (8.3)	42 (26.6)	
Poorly/Undifferentiated	258 (73.5)	29 (85.3)	91 (67.4)	22 (91.7)	116 (73.4)	
Missing	6	0	4	0	2	
Histology (*n* (%))						
Serous	284 (79.6)	17 (50)	114 (82)	20 (83.3)	133 (83.1)	<0.001 ^‡^
Other Epithelial	73 (20.4)	17 (50)	25 (18)	4 (16.7)	27 (16.9)	
Missing	0	0	0	0	0	
Cytoreduction (*n* (%))						0.09 ^‡^
R0	98 (27.8)	9 (28.1)	41 (29.9)	5 (20.8)	43 (26.9)	
Not R0	255 (72.2)	23 (71.9)	96 (70.1)	19 (79.2)	117 (73.1)	
Missing	4	2	2	0	0	
Platinum sensitivity (*n* (%))						
Sensitive	156 (55.5)	9 (37.5)	65 (55.1)	11 (68.8)	71 (57.7)	0.8 ^‡^
Resistant	125 (44.5)	15 (62.5)	53 (44.9)	5 (31.3)	52 (42.3)	
Missing	76	10	21	8	37	
Treatment Outcome (*n* (%))						
Complete Response	169 (57.5)	9 (37.5)	65 (55.1)	9 (52.9)	86 (63.7)	0.2 ^‡^
Not Complete Response	125 (42.5)	15 (62.5)	53 (44.9)	8 (47.1)	49 (36.3)	
Missing	63	10	21	7	25	

*n* may vary by characteristic due to missing data. ^†^
*p*-value was calculated using one-way ANOVA test. ^‡^
*p*-value was calculated using the chi-squared test. ^§^ Chi-squared approximation may be incorrect.

**Table 2 cancers-14-04724-t002:** Discovery cohort survival analysis.

Ovarian Cancer		Univariate Analysis (*n* = 357) ^†^			Multivariate Analysis (*n* = 342)		
Covariate	Risk Level	Hazard Ratio	95% CI	*p*-Value ^‡^	Hazard Ratio	95% CI	*p*-Value ^‡^
Age	+10 years	1.3	(1.18–1.42)	<0.001	1.32	(1.19–1.46)	<0.001
Stage	I/II/IIIA/B		Reference			Reference	
	IIIC/IV	3.45	(2.4–4.96)	<0.001	3.28	(2.05–5.26)	<0.001
Grade	Well/Moderately differentiated		Reference			Reference	
	Poorly/Undifferentiated	1.15	(0.89–1.49)	0.3	1.34	(1.02–1.76)	0.04
Histology	Other Epithelial		Reference			Stratifier ^§^	
	Serous	1.17	(0.87–1.57)	0.3			
Cytoreduction	R0		Reference			Stratifier ^§^	
	Not R0	2.43	(1.84–3.15)	<0.001			
MAGEC3 and	Loss~Exp(−)		Reference			Reference	
BRCA2	Normal~Exp(−)	1.29	(0.85–1.95)	0.2	1.14	(0.73–1.79)	0.6
	Loss~Exp(+)	1.17	(0.72–1.89)	0.5	1	(0.61–1.63)	1
	Normal~Exp(+)	1.16	(0.91–1.49)	0.2	1.14	(0.88–1.48)	0.3

^†^*n* may vary due to missing data. ^‡^
*p*-value calculated using the Wald test. ^§^ Covariate violated the proportional hazards assumption.

## Data Availability

The data and code used in this manuscript are available upon request.

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
