# Peer review of "Combined BRCA2 and MAGEC3 Expression Predict Outcome in Advanced Ovarian Cancers"

_cancers, 2022, doi:10.3390/cancers14194724_

Round 1

Reviewer 1 Report

The study entitled “Combined BRCA2 and MAGEC3 expression predict outcome in Advanced Ovarian Cancers” by Emmanuel B. Omole et al reports the potential of BRCA2 and MAGEC3 combined expression levels as prognosis predictor for ovarian cancer. The establishment of applying dual BRCA2 antibody strategy to determine BRCA protein expression by IHC can potentially be useful for future clinical biomarker exploration. The experiments and data are solid; the case selection procedure and clinical information are well-documented.  

However, the finding seems to be incremental as BRCA2 status itself is a well-established prognosis marker in ovarian cancers. The current study reported MAGEC3 can only add differential diagnosis in BRCA-expressing group will need to be validated by independent cohort.

Additional Comments:

1. Do the BRCA2 expressers vs non-expressers display differential clinical outcome in overall survival and progression-free survival?

2. Similarly, please show clinical outcome of MAGEC3 protein expression.

3. The finding should be further confirmed in an independent large cohort. 

4. Fig. 4: it will be helpful to indicate number of patients in each group.

5. Grammar and syntax errors can be identified in the manuscript. 

Reviewer 2 Report

The manuscript “Combined BRCA2 and MAGEC3 expression predict outcome in 2 Advanced Ovarian Cancers” by Omole et al. establishes that combined expression of MAGEC3 and BRCA2 may serves as a better predictor of prognosis than each marker alone. To validate the hypothesis the authors used ovarian cancer samples and did tissue microarray and immunohistochemistry to check the expression patter of protein of interest. It is well written and well-organized manuscript with some typo like H2O2 in line 144 of page 4. The few minor comments are as follows.

1.     What are the normal levels of BRAC2 and MAGEC3 in the normal ovary and how much it fluctuates during cancer condition?

2.     Did author do any quantitative analysis via ELISA etc of these two markers to establish the ratio of BRAC2:MAGEC3. Knowing the ratio in the normal condition will help to diagnose cancer condition.

Round 2

Reviewer 1 Report

The authors have adequately addressed the reviewer’s comments in scientific solidity and have improved the overall presentation of the manuscript.